# Implementation of a WSN for Environmental Monitoring: From the Base Station to the Small Sensor Node

**DOI:** 10.3390/s22207976

**Published:** 2022-10-19

**Authors:** Tiago Emanuel Oliveira, João Ricardo Reis, Rafael Ferreira Silva Caldeirinha

**Affiliations:** 1Polytechnic of Leiria, 2411-901 Leiria, Portugal; 2Instituto de Telecomunicações, 2411-901 Leiria, Portugal

**Keywords:** antenna, base station, differential, environmental, microstrip, monitoring, sensor node, WSN

## Abstract

In this paper, the implementation of a Wireless Sensor Network (WSN) for environmental monitoring (EM) is presented. It includes the design, implementation and experimental characterization of a multi-sector base station (BS) antenna composed of several microstrip Quasi-Yagi elements and the implementation and experimental characterization of a reduced form factor antenna for the sensor nodes (SN). Subsequently, it reports the implementation of a WSN based on Lopy4 transceivers, using the developed BS and SN antennas. Finally, experimental results obtained on the field to evaluate the performance of the network in terms of maximum coverage distance and coverage area are presented. According to the field tests, the connectivity between the sensor nodes and the developed WSN base station is confirmed at distances above 3.5 km and for all the antenna sectors of the multi-sector BS attaining a 360${}^{{}^{\circ}}$ of field of view.

## 1. Introduction

Wireless Sensor Networks (WSN) are often presented in the literature as a solution for large-scale autonomous monitoring systems [1,2]. They are typically composed of multiple sensor nodes (SN) scattered in a area of interest and a base station (BS). While the SN collect physical parameters of the nearby environment, the BS collects, compiles and analyzes the data from all nodes. In fact, these networks have appeared from the need of cooperative monitoring solutions, particularly aiming at the environment [3,4,5], security [6,7], health and quality of life [8,9,10,11] and agricultural [12,13] applications, among many others, or for hazard detection [14,15,16].

In particular, the authors in [3] employ a WSN in a conservation of illegal logging of the forest trees system. The foundation of this WSN is vibration sensors that are able to detect variation in the vibration frequency caused by the illegal logging. The sensors are coupled to a prototype developed based on a Zigbee platform. Whereas for the work in [8], the authors proposed a project idea for an innovative public transport system which ultimately depends on a distributed WSN for monitoring vehicle locations, traffic and environmental conditions as well as public transit supply in real time. The authors choose the multi-parameter ETL3000 station as an environmental monitoring system that measures various pollutants (i.e., CO and O${}_{3}$)and weather parameters (i.e., humidity, temperature and barometric pressure). Another example present in the literature is the work [9], where the authors present a WSN to be integrated in smart cities for waste management optimization. The authors developed a sensor node based on low energy consumption and cost, which is able to measure a trash bin filling level by ultrasounds. Finally, in [14], the authors propose Wireless Underground Sensor Networks (WUSN), to be deployed in tunnels, chambers or earth air tunnels integrated in a smart cooling system. The WUSN monitor the ambient temperature around the node; with this information, the system can then act to temperature fluctuations preventing the underground area from becoming a hazard.

In the majority of the WSN applications, including the ones reported above, the WSN benefits from having several sensors scattered in large natural areas, e.g., large portions of forest, to build a robust detection network [17]. From a practical point of view, it is then desirable that the BS is capable of covering its surroundings in an omnidirectional manner. However, antennas with such radiation patterns generally offer relatively low gain, limiting the overall dynamic range of the system. Techniques such as sectorization [18] can be employed in order to increase on the overall BS gain while providing a 360${}^{{}^{\circ}}$ field of view (FOV). This technique, illustrated in Figure 1a, aims to divide the coverage area into sectors so that higher directivity antenna can be employed. On the other hand, SN are desired to be as compact and concealed as possible, with small form factors, to reduce the visual impact when deployed in the field. Thus, to maintain the small size of the SN, the antenna size is an important factor to have in consideration at the design stage, particularly if relatively high gain antennas are required.

To this extent, project WSN-EM (PTDC/EEI-EEE/30539/2017) [19] aims to tackle part of the identified issues by developing a complete WSN for environmental monitoring and, particularly, for early stage fire detection. The project aims at the development of both BS and SN devices, including its antennas, and a novel System on the Chip (SoC) transceiver, operating the 2.4 GHz ISM band.

Thus, in line with the technical requirements of the WSN-EM project, this paper describes the implementation of a complete WSN for environmental monitoring. It is divided into several parts: firstly, it covers the fabrication and characterization of a multi-sector base station antenna composed of six microstrip waning crescent Quasi-Yagi elements. The microstrip Quasi-Yagi antenna elements, which have been already studied and optimized by the authors in [20], are experimentally characterized in a controlled laboratory environment, i.e., inside an anechoic chamber. Measurement results obtained on a physical prototype are presented and analyzed against the simulated ones of [20]. Subsequently, the impact of elements spacing on radiation pattern when disposing several Quasi-Yagi elements in a circular fashion (Figure 1b) is also evaluated. As far as the SN is concerned, the simulation results presented by the authors in [21], on a balanced slot patch antenna, are validated against experiments realized on a physical prototype of the antenna. To implement a WSN for environmental monitoring, Lopy4 transceivers are considered as an alternative to the SoC transceivers (still at the design phase), which are intended for the WSN-EM project. Finally, experimental results are realized in the field to evaluate the performance of the network in terms of maximum coverage distance and coverage area.

This paper is organized as follows: Section 2 describes the implementation of the multi-sector BS antenna composed of six waning crescent Quasi-Yagi antenna elements, including the experimental characterization realized on a prototype. In Section 3, the experimental setup for antenna measurements is being presented. This setup was used for characterizing both BS and SN antenna elements. In Section 4, details about the design and implementation of a balanced slot patch antenna for the SN, and the respective microstrip balun, are reported. Section 5 details the implementation of the WSN network based on Lopy 4 transceivers. Section 6 reports the experiments realized to evaluate the coverage area and maximum coverage distance of the proposed WSN. Finally, in Section 7, the main conclusions are drawn.

## 2. Multi-Sector Base Station Antenna

### 2.1. Microstrip Quai-Yagi Antenna Based on Waning Crescent Elements

To compose the multi-sector BS antenna architecture proposed in this paper (Figure 1b), the microstrip Quasi-Yagi antenna developed by the authors in [20] has been considered. The detailed layout of the Quasi-Yagi antenna and a photography of a produced prototype are depicted Figure 2. The antenna is based on the Quasi-Yagi principle where etched parasitic elements with a waning crescent shape are employed as the directors and reflector [22]. The antenna is then composed of a microstrip dipole as the driven element, a waning crescent reflector over a ground plane, and six optimized waning crescent directors. The waning crescent-shaped directors follow the format of a quarter of a circle centered at $x_{i}$ (Figure 2a). The frequency of operation is defined by the dimensions of the dipole arms ($d_{l}$), while the feeding line width ($f_{w}$) and length ($f_{l}$) dictate the input impedance of the antenna. The antenna was designed in a double-sided FR4 substrate with $\varepsilon_{r}=4.4$, a tanδ=0.014 and a thickness of h=1.6 mm, and it has total dimensions of 185×60 mm${}^{2}$.

According to the simulations performed in a full-wave electromagnetic solver (CST MWS), the final and optimized version of the antenna presents a realized gain of 9.4 dBi at 2.45 GHz, with a bandwidth of 450 MHz, a front-to-back ratio of 20.2 dB, and a HPBW of $63^{{}^{\circ}}$ and $54^{{}^{\circ}}$ in the azimuth and elevation planes, respectively, for the optimized dimensions of (in mm): $f_{l}=52$, g=2, $g_{w}=32$, $d_{l}=21.5$, $f_{w}=3.1$, and director dimensions of: $x_{0}=6$, $r_{0}=34.5$, $t_{0}=2.1$, $t_{1}=5$, $r_{1}=21$, $x_{1}=36$, $t_{2}=5$, $r_{2}=19$, $x_{2}=56$, $t_{3}=6$, $r_{3}=19$, $x_{3}=76$, $t_{4}=7$, $r_{4}=19$, $x_{4}=96$, $t_{5}=5$, $r_{5}=21$, $x_{5}=116$, $t_{6}=2.5$, $r_{6}=19$ and $x_{6}=136$. More details about antenna design and optimization (including parametric simulations) can be found in [20].

To further validate the model, the proposed microstrip Quai-Yagi antenna has been fabricated and experimentally characterized in laboratory environment using the experimental setup detailed in Section 3. In particular, antenna matching ($S_{11}$-parameter) and radiation patterns in the two main antenna planes (i.e., azimuth and elevation) have been measured inside an anechoic chamber and compared to the simulation results previously presented in [20]. From the $S_{11}$-parameter, depicted in Figure 3a, it is possible to observe that the measured results are in good agreement with the simulated ones, with the resonance peak present at 2.39 GHz. The antenna matching is below −10 dB between 2.15 and 2.53 GHz, presenting a bandwidth of 380 MHz, in experiments, against a slightly larger bandwidth of 450 MHz in simulations. Such discrepancy can be associated to mismatched values of the permittivity of the substrate between the real data and the one provided by the manufacturer, accuracy issues associated with the PCB production technique (produced in-house), and the use of the SMA connector, which was not considered in the simulations.

In regard to the radiation pattern obtained at 2.45 GHz, depicted in Figure 3b,c for the azimuth and elevation planes, respectively, we can observe a good agreement in the shape between the simulation and measured results. According to experiments, the antenna presents a maximum gain of of 8.9 dBi, an HPBW of 60${}^{{}^{\circ}}$ in azimuth and 54${}^{{}^{\circ}}$ elevation planes, respectively, and a back-to-front ratio of 21.2 dB. This compares to simulations by having a decrease of 0.5 dB in total gain and 1 dB in back-to-front ratio.

### 2.2. Base Station Antenna Implementation

To achieve an omnidirectional coverage, which is ideal for the proposed environmental application, the developed base station antenna is sought to be implemented in a circular disposition as already demonstrated in Figure 4a. Even though the waning crescent Quasi-Yagi antenna was originally optimized to reduce its front-to-back ratio, and the envisaged sectorization technique, performed with an RF switch, assumes that just a single antenna (sector) within the arrangement is active at a time (Time Division Multiplexing), such an antenna configuration may still be vulnerable to parasitical effects from the adjacent antennas. Thus, before final BS implementation, an additional study was performed in a simulation environment to evaluate the impact of the separation between antenna elements in the final performance. Therefore, the simulation scenario of Figure 4a was created, which is CST MWS. In particular, six antennas are equally placed over a circular shape with a radius defined by $r_{g}$, separated by 60${}^{{}^{\circ}}$, to fulfill the desired 360${}^{{}^{\circ}}$ of coverage (since the $HPBW_{az}=60^{{}^{\circ}}$ for the waning crescent Quasi-Yagi).

A parametric study on $r_{g}$ was then carried out aiming to assess the impact of adjacent antennas on the overall shape of the radiation pattern of the driven antenna. For this particular study, only the total gain and HPBW in the azimuth plane were considered. The results, depicted in Figure 4, ensure that at 2.45 GHz, the configuration with $r_{g}$ = 5 cm presents a peak gain of 8.88 dBi and an HPBW of 62.4${}^{{}^{\circ}}$; with $r_{g}$ = 6 cm, the peak gain is 8.9 dBi and the HPBW is 70.3${}^{{}^{\circ}}$; for $r_{g}$ = 7 cm, the gain increases to 9.1 dBi with an HPBW of 71.8${}^{{}^{\circ}}$; a gain of 8.5 dBi and an HPBW of 75${}^{{}^{\circ}}$ is achieved with $r_{g}$ = 8 cm. Moreover, when visually inspecting the radiation pattern’s shape, it becomes clear that the adjacent antennas act as parasitical elements, creating a small dip on the main lobe for values of $r_{g}$ inferior to 8 cm. Therefore, for the final BS implementation, a structure with $r_{g}=8$ cm was considered.

Accordingly, six equal waning crescent Quasi-Yagi antenna prototypes were constructed and displaced in a circular fashion, as depicted in Figure 5, with the assistance of a 3D-printed structure made of polylactic acid (PLA). All the fabricated antennas have been experimentally characterized to be aware of any performance discrepancy between prototypes. In order to implement the sector selection technique, a “SKY13418-485LF” RF-switch was used as a multiplexing mechanism. This 8-port switch operates from 0.1 to 6.0 GHz, and the active output is selected via digital pins (3-bits). However, as only six sectors are being considered in the proposed assembly, the remaining ports are terminated with a 50 Ω load. With the objective of characterizing the RF switch, the insertion loss and isolation between ports were measured between the input port ($P_{i}$) and all other remaining output ports ($P_{o}$). The 5 cm RF cables used between the switch and antennas are also taken into account. The results, depicted in Figure 6, are in good agreement with the data sheet given by the manufacturer with a typical isolation between ports >30 dB and average insertion loss around 1.7 dB within the 2.4 GHz ISM band.

Finally, the prototype of the base station was also experimentally characterized inside the anechoic chamber. This would allow us to assess the impact of the adjacent antenna elements in the performance of the array. Figure 7 depicts the measured $S_{11}$ and radiation pattern in the azimuth plane, for every array element, while the remaining ones are set to the OFF state (adapted at 50 Ω). According to the results, the overall $S_{11}$ of the multi-sector BS presents a useful bandwidth of 500 MHz, defined between 2.24 and 2.74 GHz, as depicted in Figure 7a, still respecting the initial project requirements. However, the radiation pattern in the azimuth plane (Figure 7b) presents a slight deformation of the main lobe, which is associated with the spacing between elements of the array, as predicted by the simulations of Figure 4. This effect could be attenuated by increasing the diameter of the circular array at the expense of undesirably increasing the volume of the antenna.

## 3. Experimental Setup for Antenna Characterization

In order to experimentally characterize the proposed antenna, both antenna matching ($S_{11}$ parameter) and radiation patterns, in the two main antenna planes, i.e., in azimuth and elevation, have been obtained. While the $S_{11}$ was measured using a two port VNA (R&S ZVM), the setup of Figure 8 was assembled inside an anechoic chamber to measure the antenna radiation patterns. The anechoic chamber was used for indoor measurements enabling experiments to be performed in a controlled, electromagnetically quiet and reflection-free radio environment.

In particular, a well-characterized *Aaronia Hyperlog 30100* antenna was used as the Tx antenna (Figure 8b). It is connected to a signal generator that generated a single tone (continuous wave) with a transmission power of 0 dBm. At the receiver end, a well-characterized *Aaronia Hyperlog 60100* antenna was connected to the spectrum analyzer to be used as reference, which was later replaced by the antenna under test (AUT). The antennas were placed 3.5 meters apart, ensuring that the measurement took place in the far-field region of the antennas. The Tx antenna was kept fixed throughout the measurements, while the AUT was rotated around its axis with the assist of a motorized pan/tilt head unit, as shown in Figure 8c. The received power was acquire utilizing a spectrum analyzer (Agilent E4408B) for each angular step with 1${}^{{}^{\circ}}$ of resolution, within the range defined between −180${}^{{}^{\circ}}$ and 180${}^{{}^{\circ}}$ (in the azimuth plane). Both Tx and Rx antennas were further rotated 90${}^{{}^{\circ}}$ to measure the radiation pattern in the elevation plane, using the same physical setup. The received power acquisition and movement control were executed in real time and post processed in MATLAB, using software developed for the effect.

## 4. Sensor Node Antenna

### 4.1. Balanced Slotted Patch Antenna

In regard to the sensor nodes of a WSN, a major requirement for such devices is the reduced weight and small form factor. Although small antennas are nowadays commercially available, e.g., chip, ceramic or Pifa antennas, those typically present very low gains (<2 dBi) and are strict to custom layout footprints.

The antenna being present in this section (Figure 9) was designed to fulfill specific project requirements, namely to achieve an antenna gain >3 dBi and have differential feeding. Hence, a small form factor balanced slotted patch antenna is proposed for the sensor node. Its layout, depicted in Figure 9a,b, was firstly introduced by Wang et al. in [23] and further optimized by the authors in [21]. The proposed design consists of a microstrip patch antenna with etched resonating slots, in both top and bottom planes, and two equidistant pins to enable differential excitation, as depicted in Figure 9. In particular, the inclusion of slots on the top patch, defined by $s_{l}\times s_{w}$, and the ones on the ground plane ($e_{l}\times e_{w}$), enable the reduction of the overall antenna dimensions while shifting down the resonant frequency of the antenna when comparing with a traditional patch antenna design. The frequency of operation can then be re-adjusted by tuning the separation between the feeding points ($f_{o}$), the dimensions of the top patch ($p_{a}\times p_{a}$) or by the distance between slots ($s_{o}$ and $b_{so}$). In particular, an antenna printed on an FR4 substrate ($\epsilon_{r}$ of 4.4 and a loss tangent of 0.014) and overall dimension of (in mm): $S_{ub}=45$, $p_{a}=22.2$, $f_{o}=5$, $s_{o}=8$, $s_{w}=0.5$, $s_{l}=15$, $e_{w}=0.7$, $e_{l}=28$, $y_{w}=22$, $y_{l}=0.4$, $r_{d}=4$ and $b_{so}=15.75$, presents according to simulations [21], a bandwidth of 130 MHz, a gain of 4.15 dBi and an HPBW of 85.5${}^{{}^{\circ}}$ on the azimuth plane. More details about the antenna design, simulation and optimization can be found in [21].

### 4.2. Balun Design

To further validate experimentally the proposed differential antenna and allow this antenna to be tested in the proposed transceiver (LoPy4), a microwave balun (balanced-unbalanced) has designed and optimized to operate at 2.45 GHz. A balun, which stands, for “balanced-to-unbalanced”, is RF device responsible to transform an unbalanced line to a balanced line and vice-versa. The proposed balun, presented in Figure 9c, is composed by a quarter-wave impedance transformer, a T-Junction power divider and a 180${}^{{}^{\circ}}$ phase shifter. For our application, the balun was designed aiming at a 50 Ω impedance in Port 1 (P1), while presenting a differential impedance of 100 Ω between Ports 2 and 3 (P2 and P3). Thus, to calculate the width of the microstrip in P2 and P3, Equation (Equation 1) is utilized [24]:
(1)$$Z_{d}\;=\;2\times Z_{0}\left(1\;-\;0.48\times e^{\left(-0.96\frac{d}{H}\right)}\right),$$
where *H* is the height of the substrate, *d* is the distance between traces (fixed at 7 mm) and $Z_{0}$ is the single-ended impedance of the microstrip, which when calculated is 50 Ω; thus, $w_{f}$ = 3.1 mm.

To attain a differential signal between P2 and P3, it is necessary that the signals reach the ports with a 180${}^{{}^{\circ}}$ phase difference. Therefore, a 180${}^{{}^{\circ}}$ phase shifter is employed, creating a shift of 180${}^{{}^{\circ}}$ by increasing the length of one of the traces by λ/2. Moreover, with the goal of splitting the power from P1 to P2 and P3 evenly, a T-junction power divided is developed. To attain a higher efficiency, the impedance of the unbalanced port was calculated by:(2)$$Z_{unb}\;=\;\frac{Z_{2}}{2}\;=\;\frac{Z_{3}}{2},$$
where $Z_{unb}$ is the impedance of the unbalanced port, while $Z_{2}$ and $Z_{3}$ represent the impedance of Ports 2 and 3, respectively (50 Ω). However, the impedance given by Equation (Equation 2) is 25 Ω, which needs to be transformed to 50 Ω to match the output line. For this reason, a quarter-wave impedance transformer was studied and added to the design. With Equation (Equation 3), it is then possible to calculate the values for wq, lf and lq.
(3)$$Z_{quarter-wave}\;=\;\sqrt{Z_{unb}\times Z_{in}},$$

The final dimensions (in mm) for the optimized balun parameters are $L_{b}=49.4$, $W_{b}=55.7$, $w_{f}=3.1$, $l_{f}=8.4$, $w_{q}=6.4$, $l_{q}=15$, $l_{a}=22.8$, $l_{1}=19.1$, $l_{2}=16.4$ and $l_{3}=15.4$.

### 4.3. Sensor Node Antenna Characterization

After proper optimization, a prototype of both antenna and the balun has been produced. A photography of the prototype is depicted in Figure 9d. Using the setup described in Section 3, the sensor node antenna was experimentally characterized. Similarly, measurements of the $S_{11}$ parameter and the radiation pattern of the prototype were obtained. Figure 10 presents measured results against the simulated ones, which were previously obtained in [21]. Both simulated and measured results are in good agreement. In experiments, the combined $S_{11}$ (balun + antenna) presents a bandwidth of 95 MHz, defined from 2.39 to 2.485 GHz, while gain is of 4.4 dBi. The antenna presents an HPBW of 84${}^{{}^{\circ}}$ and 62${}^{{}^{\circ}}$ in the azimuth and elevation planes, respectively.

## 5. WSN-EM Implementation Based on LoPy4

### 5.1. LoPy4 Transceiver Configurations

In order to implement the proposed WSN-EM architecture, LoPy4 devices [25] are considered, as they operate in the same frequency band as the envisaged WSN-EM SoC transceivers, i.e., 2.4 GHz ISM band. The LoPy4, is a micro python programmable board that can use the following communication standards: LoRa, SigFox, Wi-Fi and Bluetooth. The device incorporates an ESP32 dual core microcontroller to operate with Wi-Fi and Bluetooth communication standards, including an internal antenna (ceramic antenna) and a connector for possible external antenna integration. According to the manufacturer, utilizing a data rate of 1 Mbps and a direct-sequence spread spectrum modulation, the LoPy4 has a sensitivity that can go down to −98 dBm and a maximum output power of 20.5 dBm, with an adjacent channel rejection of 37 dB, making these devices suitable to be employed as a node or base station on a WSN.

To replicate characteristics of the project (namely the frequency of operation), Wi-Fi was chosen as the communication protocol. Therefore, the Lopy4 employed as the BS will create a Wi-Fi network, utilizing the 802.11b standard, with a channel bandwidth of 20 MHz and the Modulation Coding Scheme 0 (MSC0) [25]. With this configuration, the manufacture ensures that the Lopy4 attains a sensitivity of −93 dBm, being close to the one expected for RF-SoC to be developed in WSN-EM, which aims for a sensitivity of −92 dBm. The use of Wi-Fi opens the possibility for the creation of a user interface to control the sectors of the BS antenna or display collected data from the sensor node in a webpage, as demonstrated in Section 5.4.

### 5.2. Multi-Sector Base Station Assembly

As already indicated in Section 2, the multi-sector BS adopts a 3D-printed structure to conceal the circuitry and dispose of the antennas in the respective configuration. In addition, it employs a LoPy4 transceiver for the Wifi communication, which also controls the switching states of the RF switch, using the digital IO pins and, consequently, the active antenna sector. The RF input/output of the LoPy4 is connected to the RF switch by using a SMA to uFL pigtail. A block diagram BS configuration using the LoPy4 is as depicted in Figure 11a.

### 5.3. Sensor Node

The Lopy4 that serves as a sensor node employs a PyTrack shield that contains an integrated Global Positioning System (GPS) receiver to obtain its location. Several other sensors can be easily be connected to this expansion board. Afterwards, the node sends its current coordinates and the RSSI to the BS, which is then displayed in the user interface. The balanced microstrip slotted patch antenna is connected to the node with the balun, since the LoPy4 RF-SoC outputs an unbalanced signal, as depicted in Figure 11b.

### 5.4. User Interface

With the purpose of controlling the RF switch and the respective active BS sector and monitoring the WSN, a user interface was developed. The user interface is available only when connected to the closed network, and it can be accessed via web browser. A screenshot of the interface is depicted in Figure 12. The interface was developed in HTML and CSS, reducing the usage of the memory resource. To handle the communication between the devices, the http protocol is used. The BS creates a socket defining itself as the web server, while the remaining devices (nodes, user device) are clients. When the user creates a request in the socket (“GET”), the BS responds with a string that contains all the information for the user interface site. To change the active antenna, the user simply selects the antenna to be active and clicks the “Submit” button. The interface also informs the user of the last data received on the BS, as shown in the bottom of the site, listing the node identification, current coordinates and its RSSI. Several other pieces of information such as temperature, humidity, air pressure and others can also be added to be shown in the user interface.

## 6. WSN-EM: Field Tests

### 6.1. LoPy4 RSSI Characterization

The measurements presented in the following sections are based on the LoPy4 RSSI, which is an indicator of the received power in the device. Since RSSI is not a standardized unit of power measure, there was the need for characterization. Hence, the BS and SN LoPYs were connected back-to-back, in-line with a variable attenuator. The RSSI was then obtained for different values of attenuation. Three values were utilized for the transmission power: 2 dBm, 10 dBm and 20 dBm, while the attenuation varied from 0 to 120 dB, with a step of 10 dB. From the results, depicted in Figure 13, it is possible to observe that with a 2 dBm transmission power, the LoPy4 lost its connection with a line attenuation of 100 dB, while with 10 dBm, the link was maintained until 110 dB of attenuation. For a transmission power of 20 dBm, the LoPy4 could communicate with 110 dB, losing its bind when the line attenuation was of 120 dB.

### 6.2. Coverage Study

With the aim of validating the proposed sectorization technique, a measurement campaign was carried out, in the field, using the proposed WSN architecture. The field tests were performed in Campos do Liz, Leiria. The location is characterized by vast and large agriculture fields with short vegetation. The BS and SN were placed at a height of 1.6 m in the middle of an open field (with approx. 180 × 160 m${}^{2}$). With the BS stationary and utilizing only a single sector active for each measurement, the SN moved around the BS in random pattern, at a maximum distance of 82 m. Due to the limit area of study, the BS power was set to 2 dBm, and an attenuator of 20 dB was employed between the Lopy4 and RF switch. After connecting with the BS, the node sends its RSSI and GPS coordinates using the serial port to a computer every second. These data are then processed in real time using MATLAB. The MATLAB script associates the measured RSSI with the respective GPS coordinates obtained by the PyTrack shield, and it saves it in a file for further analysis. Figure 14 depicts the coverage results attained for sector 2 and sector 5, respectively. From the results, it can be observed that the proposed BS configuration acts as desired, covering each sector (of 60${}^{{}^{\circ}}$) when demanded.

### 6.3. Maximum Range Assessment

After studying the coverage, the possible maximum range obtained with the presented architecture was evaluated. In order to have a partial Line-of-Sight between BS and SN, an area affected by the fires of 2017 near Poço do Inglês, Marinha Grande was the chosen place for this measurement, as shown in Figure 15. This location is characterized by sand dunes with low vegetation and few remaining burned trees.

The BS was placed at an altitude of approximately 65 meters (above the sea level), while the SN, attached to the 3D-printed support (Figure 11b), was placed on a rooftop of a car (1.6 m height). The road altitude used varies from 53 to 65 m. The BS was set to have a maximum transmission power of 10 dB, respecting the limit of effective isotropic radiated power in Portugal (max. 20 dBm). Two measurements were made confirming a maximum distance of 3.85 km between the BS and sensor node, which was limited by the length of the road where the measurements were performed. In both measurement sets, 3.85 km was traveled without losing the connection established by the Wi-Fi protocol. At the farthest distance tested (3.85 km), the node presents an RSSI of −94 dBm, as depicted in Figure 16a, still ensuring Wi-Fi connectivity.

An estimation of the RSSI was also performed by utilizing the Longley–Rice radio propagation model [26] (also known as the Irregular Terrain Model (ITM)) provided in Matlab. The results of the simulation are depicted in Figure 16b for comparison. The simulations were performed by setting up the ITM model for vertical antenna polarization and maritime-over-land climate zone, while keeping the remaining model parameters as default. The gain of the BS and SN antennas was set at 8.9 dBi and 4.4 dBi, respectively, to respect the values obtained in the experimental characterization. The effective antenna height was set 2 and 1.6 m above the terrain level for the BS and SN, respectively. After analyzing the measured and estimated RSSI (Figure 16), it can be concluded that both results are in relatively good agreement. However, at approximately 3 km, the most noticeable difference occurs between the results (approx. 4 dB), which is thought to be due to the irregular terrain of the site.

## 7. Conclusions

In this paper, the implementation of a Wireless Sensor Network for environmental monitoring is being presented. It includes the design, implementation and experimental characterization of a multi-sector BS antenna based on a microstrip Quasi-Yagi with waning crescent elements. The design, implementation and experimental characterization of a reduced form factor antenna for a sensor node is also reported. The BS antenna is composed of six independent sections, controlled via an RF switch, presenting 8.9 dBi at 2.45 GHz, while the SN presents 4.4 dBi at the same frequency. Both antennas are then used with Lopy4 transceivers to create a WSN for environmental monitoring. The coverage and maximum range of the proposed architecture were also studied, where experimental results obtained in the field confirm the 360${}^{{}^{\circ}}$ FOV and the good behavior of the RF switching mechanism of the BS antenna. A maximum range of 3.85 km, limited by the measurement site location, was obtained between the SN and BS, clearly satisfying the purposes of the presented network and the abiding requirements set by the WSN-EM project.

## Figures and Tables

**Figure 1 sensors-22-07976-f001:**
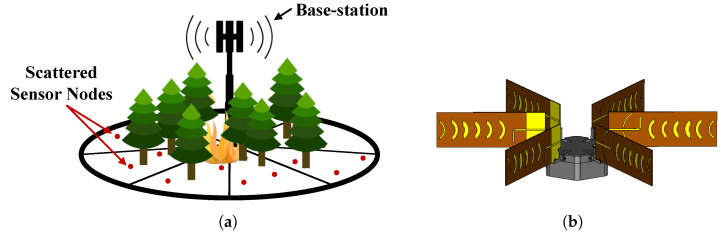
Multi-sector WSN: (**a**) Illustration of a possible configuration scenario and (**b**) proposed base station antenna.

**Figure 2 sensors-22-07976-f002:**
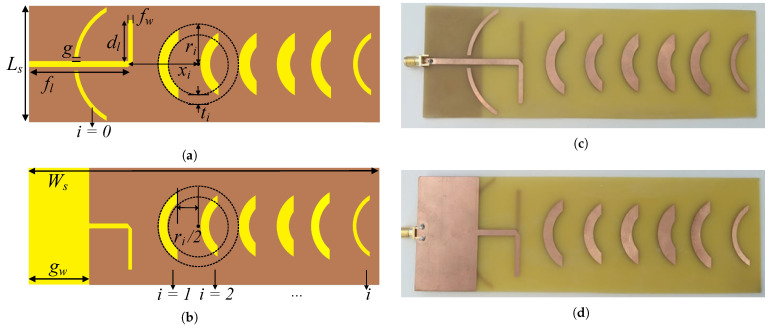
Microstrip waning crescent Quasi-Yagi: (**a**) top and (**b**) bottom antenna layout; (**c**) top and (**d**) bottom views of the antenna prototype [overall dimensions: 185×60 mm${}^{2}$].

**Figure 3 sensors-22-07976-f003:**
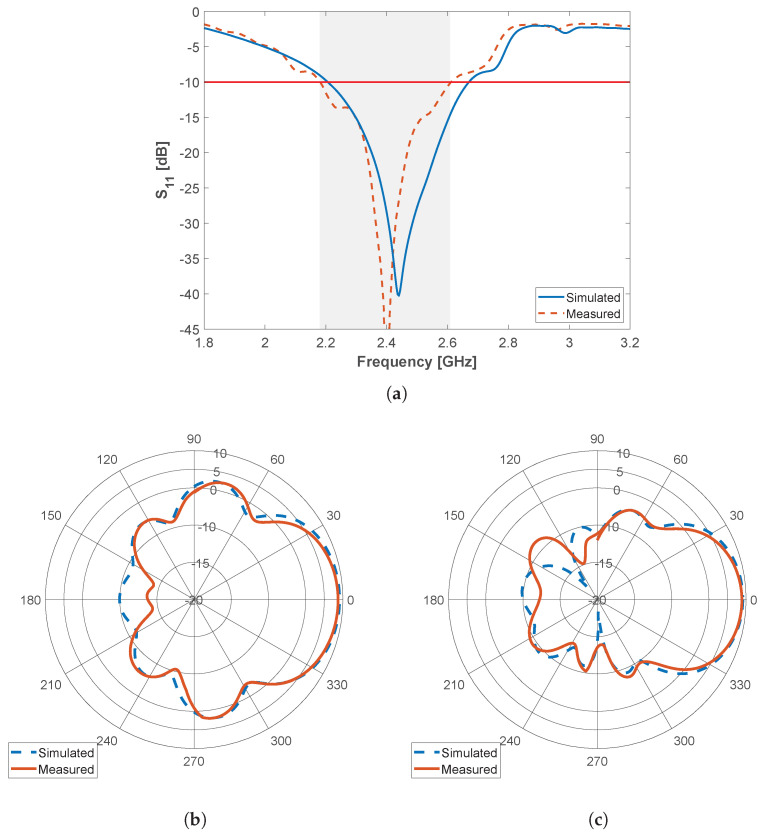
Simulated and measured results for the waning crescent Quasi-Yagi antenna element: (**a**) $S_{11}$-parameter and radiation patterns at 2.45 GHz in the (**b**) azimuth and (**c**) elevation plane.

**Figure 4 sensors-22-07976-f004:**
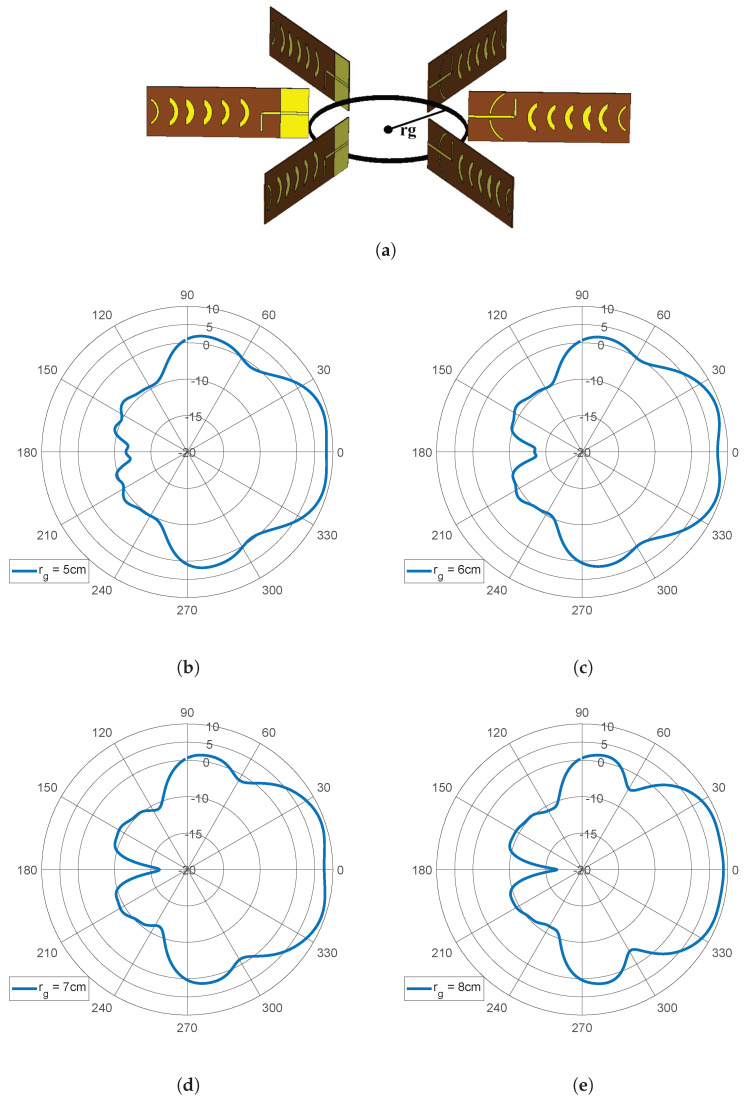
Study of the impact of the adjacent antennas in radiation performance: (**a**) simulation model in CST MWS; simulated radiation patterns in azimuth plane for: (**b**) $r_{g}$ = 5 cm, (**c**) $r_{g}$ = 6 cm, (**d**) $r_{g}$ = 7 cm and (**e**) $r_{g}$ = 8 cm.

**Figure 5 sensors-22-07976-f005:**
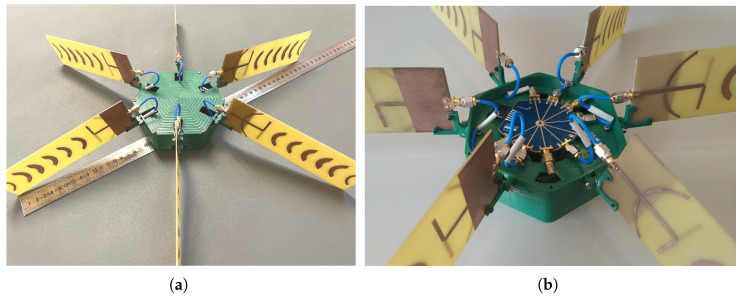
Multi-sector BS: (**a**) complete antenna prototype and (**b**) open view of the SKY13418-485LF RF switch.

**Figure 6 sensors-22-07976-f006:**
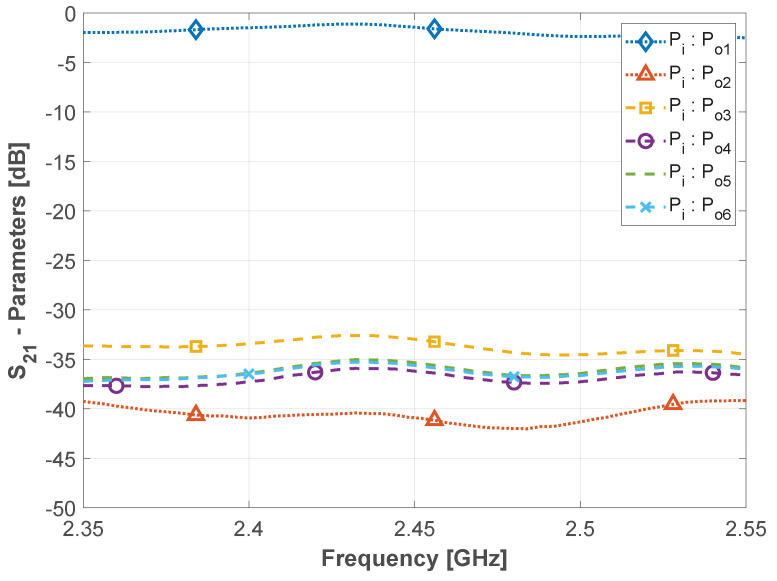
Measured isolation and insertion loss of the RF switch (plus cables).

**Figure 7 sensors-22-07976-f007:**
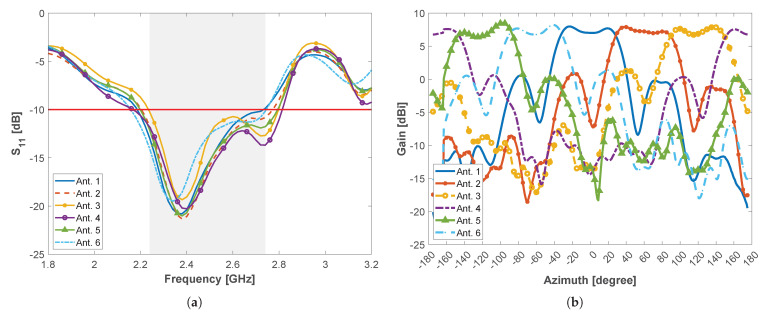
Multi-sector BS antenna: (**a**) $S_{11}$-parameter and (**b**) radiation pattern in the azimuth plane, per antenna element, considering the remaing ports in the OFF state.

**Figure 8 sensors-22-07976-f008:**
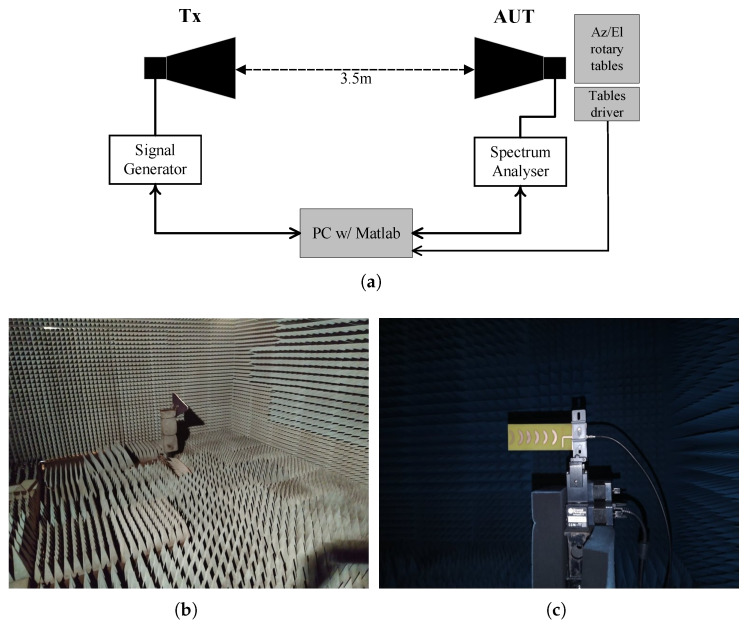
Radiation pattern measurement setup: (**a**) block diagram; (**b**) photography of Tx antenna and (**c**) close up an AUT, inside the anechoic chamber.

**Figure 9 sensors-22-07976-f009:**
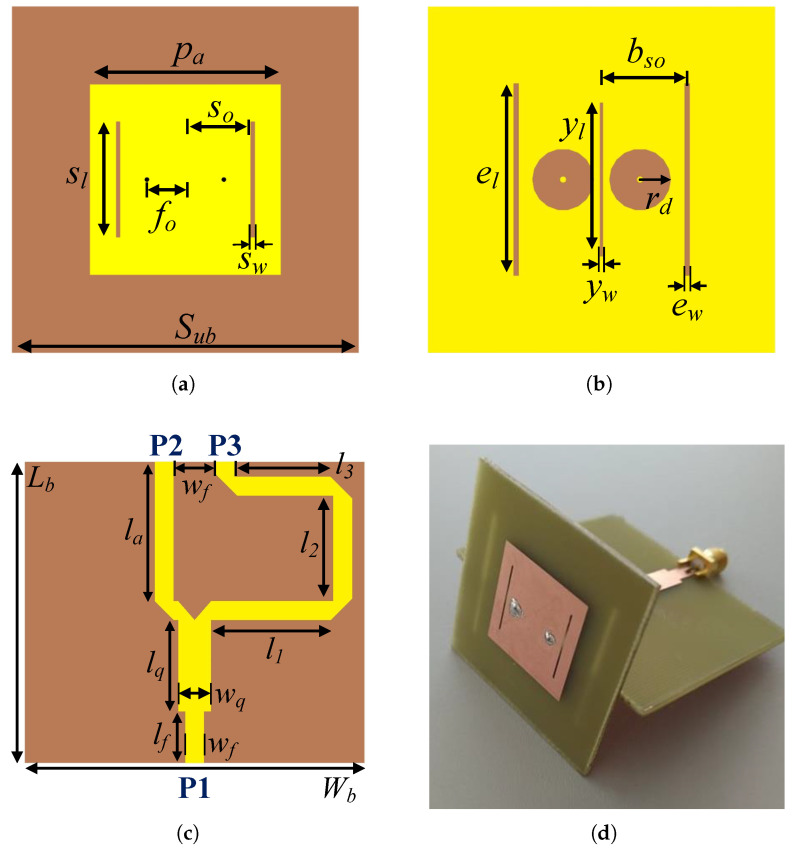
Sensor node antenna: (**a**,**b**) top and bottom layout views, respectively; (**c**) balun top view, and (**d**) final antenna prototype (overall dimensions: 55×55×50 mm${}^{3}$).

**Figure 10 sensors-22-07976-f010:**
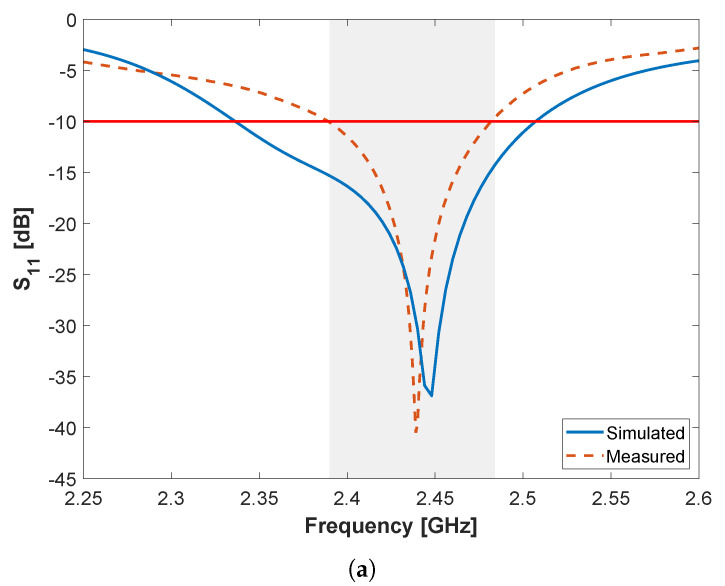
Measured and simulated (**a**) $S_{11}$ parameter and radiation pattern in: (**b**) the azimuth (**c**) elevation planes, respectively, for the balanced microstrip slotted patch antenna (including the balun).

**Figure 11 sensors-22-07976-f011:**
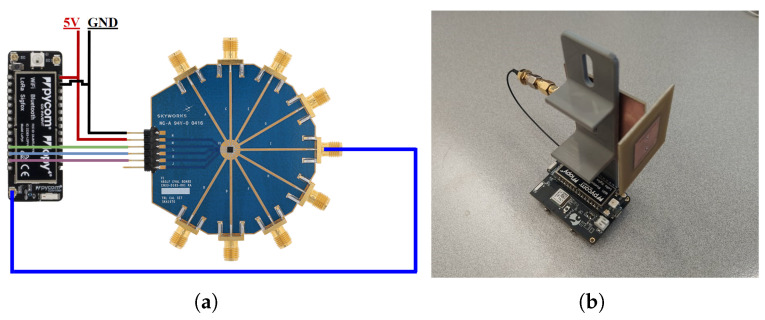
(**a**) Block diagram of BS circuitry and (**b**) photography of the SN.

**Figure 12 sensors-22-07976-f012:**
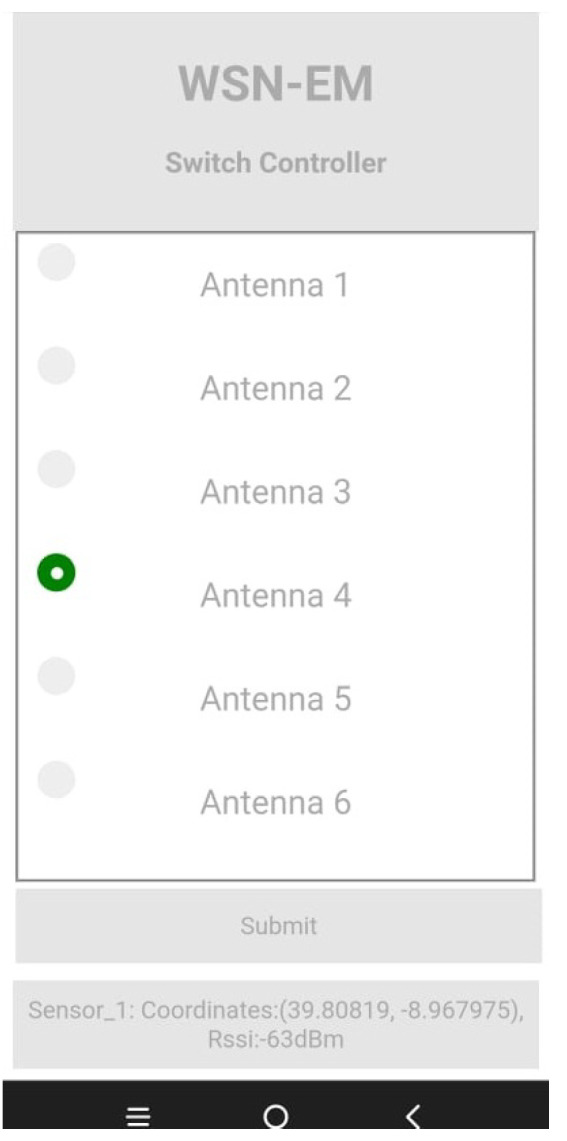
Screenshot of the user interface.

**Figure 13 sensors-22-07976-f013:**
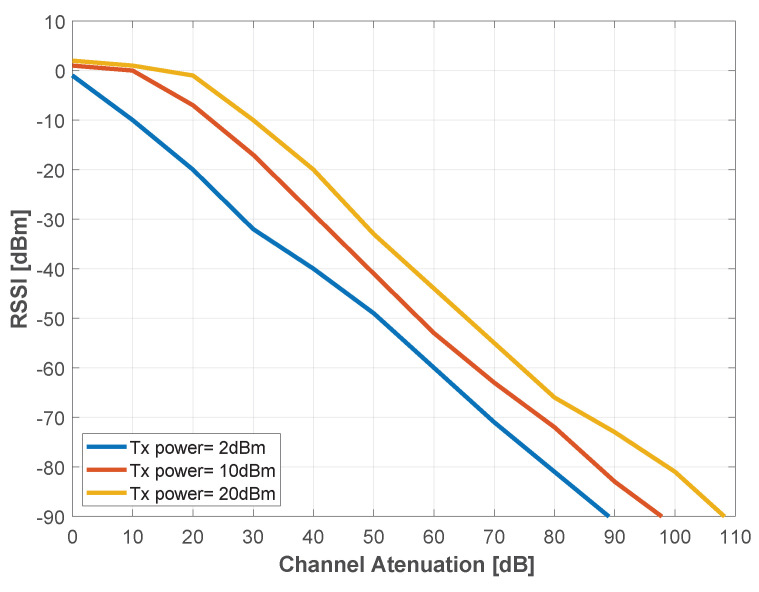
Experimental LoPy4 RSSI characterization.

**Figure 14 sensors-22-07976-f014:**
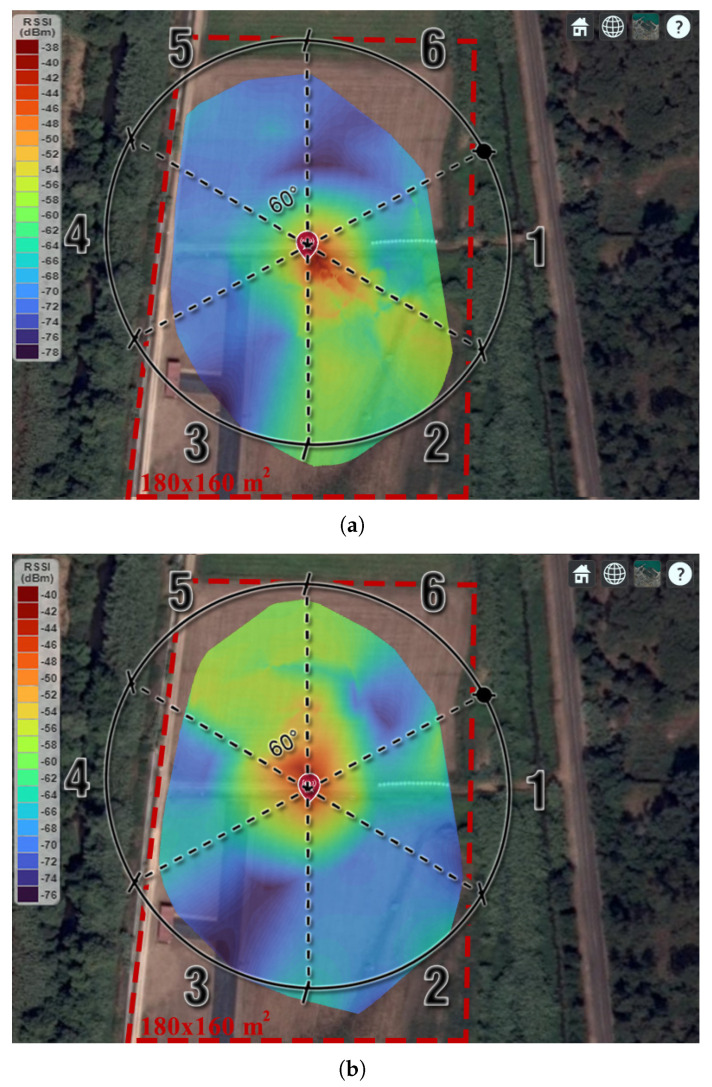
Field test results for the BS coverage campaign with: (**a**) Sector 2 and (**b**) Sector 5 active.

**Figure 15 sensors-22-07976-f015:**
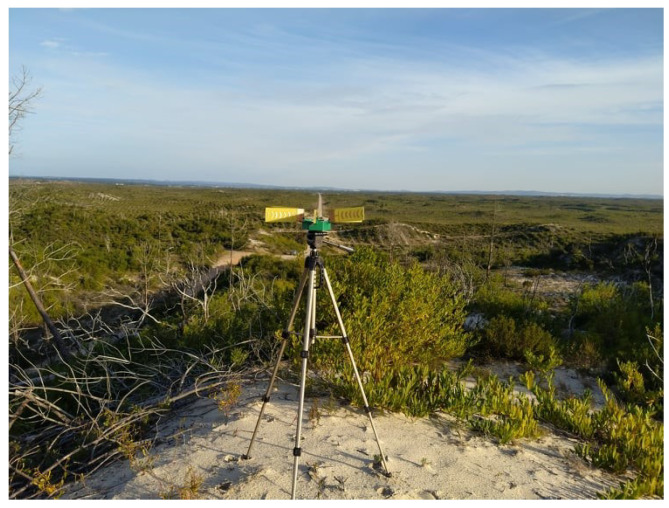
Photography of the BS placement on the WSN range campaign in Poço do Inglês, Marinha Grande.

**Figure 16 sensors-22-07976-f016:**
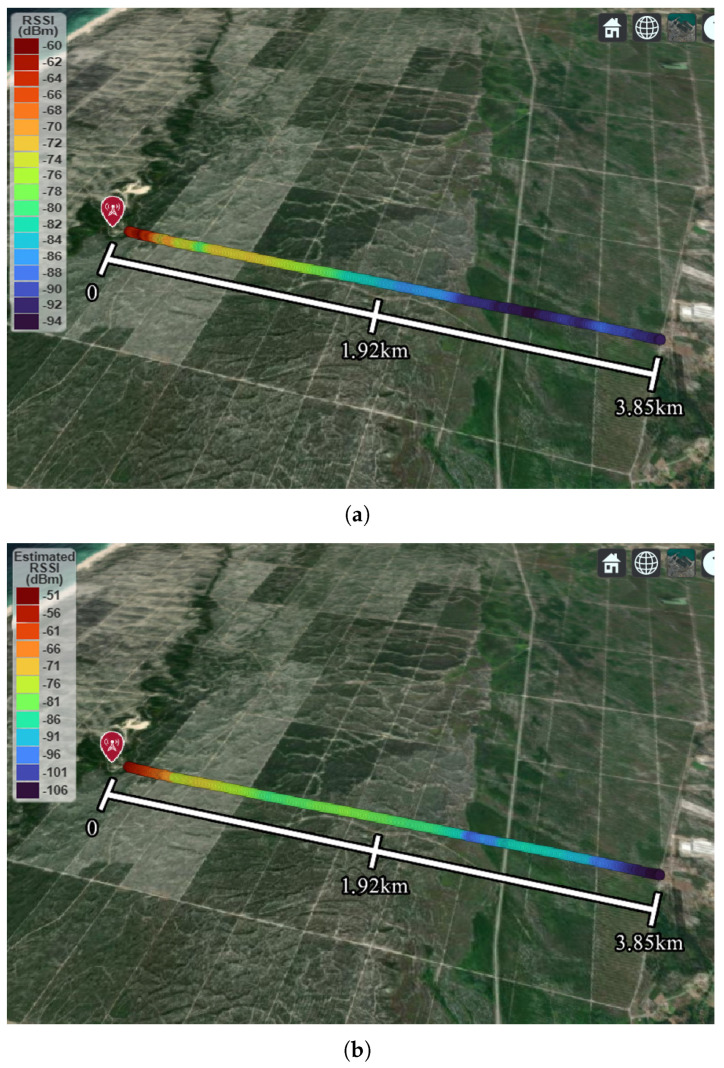
Maximum range campaign: (**a**) measured RSSI and (**b**) estimated RSSI using ITM radio propagation model of the proposed WSN.

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
