# Peer review of "Implementation of a WSN for Environmental Monitoring: From the Base Station to the Small Sensor Node"

_sensors, 2022, doi:10.3390/s22207976_

Round 1

Reviewer 1 Report

In this paper the authors implemented a wireless sensor network (WSN) with their proprietary antenna design. The authors conducted the research with solid design and test process and finally validated the system with field test.

However, I have the following concerns which may provide some ideas to improve the quality of this paper:

1) In Fig. 2 (a) and (b), it is hard to read the text on the figures because the size is too small. 

2) In Fig. 2 and Fig. 8, it is better to specifify some size information about the antennas so that the readers can validate the characteristics of the antennas.

3) Similarily, it is better to provide some size reference beside the multi-sector BS antenna in Fig. 5 so that the readers can know about the size of the antenna at first glance.

4) In Fig. 13, it is better to specify on the figure where is the target 180m x 160m open field.

Author Response

Dear Reviwer, 

Please find our point-by-point reply to your comments in the attached PDF file. 

Best Regards, 

João Reis

Reviewer 2 Report

This paper proposes a WSN based on the multisector antenna as BS and a compact antenna as SN. The paper investigates the all step of design, fabrication and test, step by step in as an engineering repot. however some comments must be mentioned:

1- The novelty of work is poor. The both of BS and SN antenna have been proposed in previous works. However integration of the all parts in a WSN can be exciting. 

2- The results of yagi antenna (S11 and patterns) must be given in array case (case of Fig. 5), where the other antennas in BS are off. I think the parasitic elements (the other 5 yagi antennas) has an effect on driven yagi antenna, especially in s11 and pattern.

3-The insertion loss of switch is so high in ISM band, it is not a good switch for this application. 

4-I think the information in the third part of the article is not useful. It is suggested that this section be completely deleted.

5- The friis equation (Eq. 4) is an ideal estimation only for free space loss and is not a good estimation in this work. the multipath fading and environments effects have been not include in this equation. 

Author Response

Dear Reviwer, 

Please our point-by-point reply to your comments in the attached PDF.

Many thanks for your inputs.

King regards, 

Joao Reis 

Round 2

Reviewer 2 Report

The my comments have been answered in the revised version.